# Dominant CT Patterns and Immune Responses during the Early Infection Phases of Different SARS-CoV-2 Variants

**DOI:** 10.3390/v15061304

**Published:** 2023-05-31

**Authors:** Kentaro Nagaoka, Hitoshi Kawasuji, Yusuke Takegoshi, Yushi Murai, Makito Kaneda, Kou Kimoto, Hideki Tani, Hideki Niimi, Yoshitomo Morinaga, Kyo Noguchi, Yoshihiro Yamamoto

**Affiliations:** 1Department of Clinical Infectious Diseases, Toyama University Graduate School of Medicine and Pharmaceutical Sciences, 2630 Sugitani, Toyama 930-0194, Japan; 2Clinical and Research Center for Infectious Diseases, Toyama University Hospital, 2630 Sugitani, Toyama 930-0194, Japan; 3Department of Virology, Toyama Institute of Health, 17-1 Nakataikouyama, Imizu-shi 939-0363, Japan; 4Department of Clinical Laboratory and Molecular Pathology, Toyama University Graduate School of Medicine and Pharmaceutical Sciences, 2630 Sugitani, Toyama 930-0194, Japan; 5Center for Advanced Antibody Drug Development, Toyama University Graduate School of Medicine and Pharmaceutical Sciences, 2630 Sugitani, Toyama 930-0194, Japan; 6Department of Microbiology, Toyama University Graduate School of Medicine and Pharmaceutical Sciences, 2630 Sugitani, Toyama 930-0194, Japan; 7Department of Radiology, Toyama University Graduate School of Medicine and Pharmaceutical Sciences, 2630 Sugitani, Toyama 930-0194, Japan

**Keywords:** pneumonia, COVID-19, CT, ground-glass opacity, Omicron, organizing pneumonia

## Abstract

Ground-glass opacity (GGO) and organizing pneumonia (OP) are dominant pulmonary CT lesions associated with COVID-19. However, the role of different immune responses in these CT patterns remains unclear, particularly following the emergence of the Omicron variant. In this prospective observational study, we recruited patients hospitalized with COVID-19, before and after the emergence of Omicron variants. Semi-quantitative CT scores and dominant CT patterns were retrospectively determined for all patients within five days of symptom onset. Serum levels of IFN-α, IL-6, CXCL10, and VEGF were assessed using ELISA. Serum-neutralizing activity was measured using a pseudovirus assay. We enrolled 48 patients with Omicron variants and 137 with precedent variants. While the frequency of GGO patterns was similar between the two groups, the OP pattern was significantly more frequent in patients with precedent variants. In patients with precedent variants, IFN-α and CXCL10 levels were strongly correlated with GGO, whereas neutralizing activity and VEGF were correlated with OP. The correlation between IFN-α levels and CT scores was lower in patients with Omicron than in those with precedent variants. Compared to preceding variants, infection with the Omicron variant is characterized by a less frequent OP pattern and a weaker correlation between serum IFN-α and CT scores.

## 1. Introduction

Coronavirus disease 2019 (COVID-19) is a highly transmissible disease caused by severe acute respiratory syndrome coronavirus 2 (SARS-CoV-2), which has infected over 766 million individuals worldwide since the beginning of the COVID-19 pandemic [1,2]. This pandemic has affected the capacity of local and regional healthcare systems worldwide, resulting in temporal exhaustion of in-hospital medical services and increasing diagnostic delay [3,4]. By late 2021, the Omicron variant rapidly outcompeted the Delta variant and dominated the pandemic [5]. Although Omicron has higher transmissibility, even to individuals who had received the vaccine [6], it is regarded as less virulent in terms of the rate of hospitalization, incidence of severe disease, and mortality [7,8]. The reduced clinical severity of the Omicron variant has been partly attributed to intrinsic viral factors, as suggested by the inefficient replication of Omicron in human alveolar organoids and ex vivo-infected lung tissues [9] and by its increased resistance toward innate immune defenses, including type I interferons (IFNs) [10,11].

Similarly, the Omicron variant is associated with fewer and less severe changes in chest computed tomography (CT) images than the Delta or precedent variants [12,13]. For the Delta and precedent variants, the typical appearance of COVID-19 pneumonia on chest CT scans of patients with moderate to severe disease involves ground-glass opacity (GGO) [14,15]. Organizing pneumonia (OP)-like lesions are a dominant feature in patients with mild COVID-19 [14]. Thus, a scoring system based on the extent and density of pulmonary inflammatory lesions, predominantly GGOs, is a reliable approach for predicting the clinical severity of COVID-19 [16,17]. However, it remains unclear how different immune responses against SARS-CoV-2 are related to dominant CT patterns and how these relationships have changed after the emergence of the Omicron variant.

Here, we hypothesized that different early immune responses against SARS-CoV-2 variants are reflected in CT patterns of GGO and OP. To elucidate the pathophysiological implications of CT findings among the different SARS-CoV-2 variants, we assessed serum levels of IFN-α (type I IFNs), IL-6, CXCL10, and VEGF, which are representative of the innate immune response [18,19,20,21], as well as serum-neutralizing activity, which reflects humoral immune responses against SARS-CoV-2 infection and is used as an indicator of vaccine efficacy [22,23].

The primary objective of this study is to evaluate the correlations between immune indicators and dominant CT patterns during the early phase of SARS-CoV-2 infection and how these correlations change between the Omicron variant and precedent variants.

## 2. Materials and Methods

### 2.1. Study Design

This study was conducted as part of the Toyama University COVID-19 Cohort Study, an investigator-initiated prospective single-center study designed primarily to investigate the clinical, epidemiological, radiological, and microbiological features of COVID-19. Participants were diagnosed with COVID-19 based on the findings of reverse transcription quantitative polymerase chain reaction (RT-qPCR) assays. Nasal specimens for RT-qPCR were collected, and chest CTs were performed upon hospital admission. Serum samples were stored at −80 °C following each laboratory examination. This study was approved by the Ethical Review Board of the University of Toyama (R2019167), and written informed consent was obtained from all participants (except for three participants, whose informed consent was obtained from the participants’ next of kin).

The study period was between December 2020 and April 2022, which covered four major waves of the pandemic in Japan: third wave (December to January 2021), fourth wave (April to June 2021, mainly attributed to the Alpha variant), fifth wave (July to October 2021, mainly attributed to the Delta variant), and sixth wave (January to April 2022, mainly attributed to the Omicron BA.1 variant).

The inclusion criteria were as follows: (1) age 18 years or older, (2) hospitalization at our hospital during the study period, and (3) blood samples collected within five days after symptom onset. Participants who had participated in other clinical trials were excluded from the study.

### 2.2. Study Participants and Study Protocol

Since the first monovalent mRNA vaccine against SARS-CoV-2 wild-type (WT), which had less efficacy against the Omicron variant, became widely available during the fifth wave in Japan, few inpatients received the vaccine until the fifth wave, whereas a large number of inpatients received the vaccine twice before admission during the sixth wave. Therefore, we included the following participants for further analysis (Figure 1): those who had not received the vaccine in the third to fifth waves (Delta and precedent variants), and those who had received the vaccine twice at least two weeks before admission in the sixth wave (Omicron BA.1 after vaccine).

Data on participant demographics, comorbidities, clinical presentation, laboratory findings, therapy regimen, and prognosis were collected from participants’ medical charts.

Hypoxemia requiring oxygen therapy was defined as a blood oxygen saturation (SpO2) level of ≤93% at rest/motion in room air, as defined previously [24].

### 2.3. Image Analysis and Classification of CT Patterns

Chest CTs were performed using a multidetector CT scanner (SOMATOM Definition AS+; Siemens Healthineers, Tokyo, Japan) and SOMATOM Go. Top (Siemens Healthineers). Scanning parameters were identical to the manufacturer’s standard recommended presetting for the thoracic routine. Images were reconstructed using a 1 mm slice thickness and a high spatial resolution algorithm. All participants underwent CT scanning of the chest in the supine position during end inspiration.

Two experienced pulmonary radiologists (KN and KN) with >19 years of experience reviewed previous chest CT scans and categorized the radiological findings according to the Fleischner Society Glossary of Terms for Thoracic Imaging [25]. When a newly developed inflammatory lesion was detected by chest CT performed on admission, COVID-19 pneumonia was subsequently confirmed [26,27]. In this study, we classified the dominant CT patterns into GGO and OP patterns. Scans that contained both GGO and OP were assigned to that of the dominant pattern. CT patterns that did not correspond to either pattern were categorized as other. Both radiologists reached a consensus on the classification of radiological manifestations.

### 2.4. Chest CT Score

Semi-quantitative CT severity scores were calculated per lobe for each of the five lobes, considering the extent of anatomic involvement, in accordance with a previous report [16]: 0, no involvement; 1, <5% involvement; 2, 5–25% involvement; 3, 26–50% involvement; 4, 51–75% involvement; and 5, >75% involvement.

### 2.5. Cytokine Measurement

Serum cytokines and chemokines (IFN-α, IL-6, CXCL10, and VEGF) were measured using commercially available enzyme-linked immunosorbent assay (ELISA) kits, according to the manufacturers’ instructions. IFN-α levels were measured using the VeriKine-HS Human IFN Alpha All Subtype ELISA Kit (PBL Assay Science, Piscataway, NJ, USA), CXCL10 levels were measured using the Human CXCL10/IP-10 ELISA Kit (Proteintech, Rosemont, IL, USA), IL-6 levels were measured using the AuthentiKine™ Human IL-6 ELISA Kit (Proteintech, Rosemont, IL, USA), and VEGF levels were measured using the AuthentiKine™ Human VEGF ELISA Kit (Proteintech, Rosemont, IL, USA). Each sample was measured at first sight. If an analyte signal was below the background signal, it was set to zero; if the signal was detectable but below the manufacturer’s lower limit of quantification, it was set to the lower limit of detection.

### 2.6. RT-qPCR

RT-qPCR (for detecting SARS-CoV-2) was performed as previously described [28]. Quantification quality was controlled using the AcroMetrix COVID-19 RNA Control (Thermo Fisher Scientific, Waltham, MA, USA). The detection limit was approximately 0.4 copies/μL (2 copies/5 μL). RNAemia was determined when SARS-CoV-2 was detectable in the blood serum specimens.

### 2.7. Pseudovirus Neutralization Assay

The neutralizing activity of human serum against pseudoviruses was measured using a high-throughput chemiluminescent reduction-neutralizing test, as previously described [23,29]. Briefly, Vero cells (E6/TMPRSS2) were treated with 100-fold dilutions of sera from patients and were then inoculated with pseudo-type SARS-CoV-2. The infectivity of the pseudoviruses was determined by measuring luciferase activity after 24 h of incubation at 37 °C and was expressed as the mean of duplicate measurements. The values for the samples without pseudovirus and with pseudovirus but without serum were defined as 0% and 100% infection (100% and 0% inhibition), respectively. To measure the neutralizing activity against the infected variant of each pandemic wave, we used four pseudoviruses with expression plasmids for the truncated S protein of SARS-CoV-2: pCAG-SARS-CoV-2 S (Wuhan; WT); pCAGG-pm3-SARS2-Shu-d19-B1.1.7 (Alpha-derived variant); pCAGG-pm3-SARS2-Shu-d19-B1.617.2 (Delta-derived variant); and pCAGG-pm3-SARS2-Shu-d19-B1.1.529.1 (Omicron BA.1-derived variant) [23,29].

### 2.8. Statistical Analysis

The participants’ medical and demographic characteristics were summarized using medians (interquartile ranges) or numbers (percentages). Differences between two groups were tested using the Mann–Whitney U test or Fisher’s exact test. The Mann–Whitney U test with Bonferroni correction was used to compare nominal variables among three groups. Spearman’s rho correlation coefficients were estimated for all pairs of immune parameters and viral loads. The results of the correlations between immune parameters were summarized in a correlation matrix. The size of the tests was set to 0.05, and statistical significance was set to *p* < 0.05. JMP Pro 16 (SAS Institute, Cary, NC, USA) and GraphPad Prism 9 software (GraphPad Software, San Diego, CA, USA) were used for statistical analyses.

## 3. Results

### 3.1. Clinical Features of Participants in This Study

The clinical features of the participants are summarized in Table 1. After exclusion, 137 participants with Delta and precedent variants (39 in the third wave, 50 in the fourth wave, and 48 in the fifth wave) and 48 participants with the Omicron variant after vaccination were included for further analysis. Part of the study population during the third to fifth waves was included in a previous report that investigated the association between subpleural GGO, respiratory failure, and viremia [30]. Amongst the study population, one patient developed COVID-19 in the hospital (incidental COVID-19), contracting the Delta variant. Age, underlying diseases, BMI, and CRP levels differed between participants with the Delta and precedent variants and those with the Omicron variant (*p* < 0.05, respectively).

### 3.2. Radiological Findings

The radiological findings are summarized in Table 2. None of the participants underwent CT after the initiation of therapy. The CT evaluation day after clinical onset was earlier in participants with the Delta and precedent variants than in those with Omicron (3 (2–4] vs. 2 (1–3], *p* < 0.005). Pulmonary lesions were absent in 67% of the participants with Omicron, whereas certain pulmonary lesions were present in 55% of the participants with the Delta and precedent variants. Semi-quantitative CT scoring revealed that the CT scores were higher in participants with the Delta and precedent variants than in those with Omicron (*p* < 0.05). A clear correlation in CT scores was confirmed between the two radiologists (R = 0.984, *p* < 0.001). Notably, the occurrence of OP patterns was significantly higher in participants with the Delta and precedent variants than in those with Omicron (20% vs. 2%, *p* < 0.01). Other CT findings, including GGO patterns, were similar between participants with Omicron and precedent variants.

### 3.3. Serum Immune Indicators and Dominant CT Patterns at the Early Phase of SARS-CoV-2 Infection

The results of the immune indicator-level analyses related to dominant CT patterns are summarized in Figure 2 and Figure 3. Among participants with the Delta and precedent variants, age and IL-6 levels were higher in participants with GGO or OP than in those without pulmonary lesions (*p* < 0.01, respectively) (Figure 2). IFN-α and CXCL10 were higher in participants with GGO than in those without pulmonary lesions (*p* < 0.01, respectively). Neutralizing activity and VEGF levels were higher in participants with OP than in those without pulmonary lesions (*p* < 0.05, respectively). Participants with GGO and those with OP were different ages (GGO vs. OP vs. absence of pulmonary lesions, 53 (49–58] vs. 40 (34–51] vs. 35 (24–51]).

Among participants with the Omicron variant, age and IL-6 levels were higher in those with GGO than in those without pulmonary lesions (*p* < 0.05, respectively) (Figure 3). As OP-like lesions were observed in only one participant with the Omicron variant, we could not assess the differences in immune indicator levels between the dominant CT patterns.

### 3.4. Correlations among CT Score and Immune Indicator Levels

In participants with the Delta and precedent variants, the CT score was directly correlated with age and all evaluated immune indicators (Figure 4A), with the strongest correlation observed for IL-6 (r = 0.66), followed by age (r = 0.43), IFN-α (r = 0.37), and CXCL10 (r = 0.32) (*p* < 0.0001).

In participants with the Omicron variant, the CT score directly correlated with age and all evaluated immune indicators other than IFN-α and neutralizing activity (Figure 4B), with the strongest correlation observed for IL-6 (r = 0.47; *p* < 0.0001), followed by age (r = 0.38; *p* < 0.0001), CXCL10 (r = 0.34; *p* < 0.05), and VEGF (r = 0.32; *p* < 0.05). IFN-α levels did not appear to be correlated with age or IL-6 levels (vs. age, r = 0.09; *p* = 0.52, vs. IL-6, r = 0.09; *p* = 0.55).

The landscape of immune response and dominant CT patterns during the early phase of SARS-CoV-2 infection are summarized in Figure 5.

### 3.5. Serum Immune Indicator Levels and CT Findings in Unvaccinated Participants with the Omicron Variant

To confirm the impact of the vaccine on the early immune response against the Omicron variant, we also assessed the immune indicator levels of participants with COVID-19 in the sixth wave who had not received any vaccine against SARS-CoV-2 before infection (n = 24; Appendix A). As shown in Appendix A, pulmonary lesions were absent in 46% of unvaccinated participants with Omicron, and OP-like lesions were observed in only one participant (4%).

The correlations between immune indicator levels and CT pattern were similar to those of vaccinated participants with the Omicron variant (Appendix A). That is, the CT score was directly correlated with age, IL-6, and CXCL10, with the strongest correlation observed for CXCL10 (r = 0.59, *p* < 0.005), followed by IL-6 (r = 0.48; *p* < 0.05), and age (r = 0.46; *p* <0.05). Similarly, IFN-α levels did not appear to be correlated with age or IL-6 levels (vs. age, r = −0.11; *p* = 0.62, vs. IL-6, r = 0.19; *p* = 0.38).

## 4. Discussion

In this study, we demonstrated that early COVID-19 immune responses differed between participants with GGO and those with OP for infections with the Delta and precedent variants. Specifically, serum IFN-α and CXCL10 levels were strongly associated with the presence of GGO, whereas neutralizing activity and VEGF were associated with OP. Moreover, the participants with GGO as the dominant CT pattern were significantly older than those with OP as the dominant CT pattern. However, for infection with the Omicron variant, the prevalence of the OP pattern was significantly decreased, and the correlation between the CT score and IFN-α was weaker. In contrast, IL-6 showed a strong correlation with CT score for all participants with COVID-19 in this study, regardless of the variant.

In terms of the correlation between radiological findings and pathology, GGO in moderate to severe COVID-19 typically corresponds to histopathological findings, such as diffuse alveolar damage or acute fibrinous organizing pneumonia with or without vascular damage and thrombosis [31]. OP reflects histopathological findings in milder COVID-19 cases, such as type 2 pneumocyte hyperplasia, interstitial inflammation, intra-alveolar edema with proteinaceous exudates, and organization without fibrin or hyaline membranes [32,33]. However, efforts to elucidate the mechanisms of these radiology–pathology changes have seldom investigated the association between immune responses and CT findings.

To the best of our knowledge, this study is the first to demonstrate the differences in immune characteristics between participants with GGO and those with OP in COVID-19. Higher neutralizing activity in participants with the OP pattern, rather than a stronger innate immune response (IFN-α and CXCL10), is considered to be consistent with the better prognosis of OP because an earlier humoral immune response has previously been linked to a better prognosis of COVID-19 [34,35].

In this study, we speculate that the intrinsic viral factors, rather than the impact of the vaccine, predominantly affected the CT patterns of participants with the Omicron variant. This is because the history of vaccination (vaccinated twice or unvaccinated) was less associated with the early immune response and CT patterns in the participants from the sixth wave. As a possible mechanism, the Omicron variant first proliferates rapidly in the upper airway because of increased resistance to the innate immune defense, IFNs [10,11]. In the upper airway, the nasal cavity is the location where the highest multiplicity of infection due to SARS-CoV-2 viruses per unit tissue surface area occurs, and is associated with aspecific physiological defenses [36,37]. After infection in the upper airway, owing to inefficient replication in human lungs [10,11], proliferation or invasion of the Omicron variant may be highly suppressed when the immune response or aspecific physiological defenses have been effectively induced. When a less effective immune response has been induced after infection in the upper airway, proliferation or invasion of the Omicron variant in the lungs might be achieved, which mostly reflected as pulmonary GGO lesions. In this study, we could not elucidate the reason why the OP pattern decreased in patients with the Omicron variant. Since the incidence of the OP pattern in COVID-19 with the Omicron variant is not fully known, further investigation might be necessary to understand the association between the OP pattern and the Omicron variant.

A previous report showed that IFN-α serum levels in COVID-19, prior to the emergence of the Omicron variant, reflect the systemic immune response against SARS-CoV-2 invasion into pulmonary circulation because they are significantly associated with the presence of pneumonia and viremia [38]. Accordingly, we speculated that the attenuated invasion of Omicron into the pulmonary circulation may result in a lower correlation between IFN-α and CT scores. Recent reports have focused on the role of IFNs in Omicron infection as the causative factor for increased transmissibility or as potential therapeutic targets [10,39]. Further investigation is required to elucidate how Omicron infections affect the innate immune response via IFN-α production during the development of pneumonia.

In this study, IL-6 was significantly associated with the presence of GGO and CT scores in all participants with COVID-19, which agrees with previous studies demonstrating a significant predictive value of IL-6 in relation to CT scores for the prognosis of COVID-19 [40,41]. Moreover, our study showed that the association between CT scores and IL-6 did not change with the Omicron variant. According to the findings of our study, we believe that the immune pathway related to IL-6 plays a central role in the development of COVID-19 pneumonia, even in the Omicron variant, which supports the application of immunosuppressant therapies in SARS-CoV-2 infection, including tocilizumab [18]. To date, antiviral drugs, as well as adequate vaccines, have been authorized and recommended for high-risk COVID-19 patients with the Omicron variant to prevent hospital admission and death [42]. IL-6 elevation with the presence of GGO and a high CT score might be indicative for additional intervention with immunosuppressant therapy in patients with the Omicron variant. In this study cohort, we experienced one patient with COVID-19 due to Omicron variant who presented prolonged IL-6 elevation with deteriorated pulmonary lesions and subsequently required > 2 months of corticosteroid administration. The prolonged consequences of COVID-19 are known as long COVID-19, and dyspnea is a relatively large part of those [43]. Based on this, we consider the possibility that IL-6 elevation with the presence of GGO in the early phase of COVID-19 might also associate with the development of long COVID-19. Further research would be necessary focusing on early IL-6 values in COVID-19 pneumonia.

Nevertheless, this study has several limitations. First, the single-center observational study design may have resulted in selection bias. Secondly, we could not identify the causative strain in the third pandemic wave because genetic identification of the epidemic strain was not routinely conducted at that time. Therefore, our results for participants from the third wave (n = 39) did not completely reflect the neutralizing activity against the infected strain. However, considering our consistent results and the associations detected between immune indicators and radiological findings, we believe that these limitations did not significantly affect our findings.

## 5. Conclusions

We demonstrated that the early immune response against SARS-CoV-2 differs among COVID-19 variants and may be reflected by different CT findings, including the presence of GGO and OP. Analysis of participants infected with the Omicron variant revealed a decrease in the OP pattern and a weaker correlation between serum IFN-α levels and CT severity scores, which provides important insights for further research on SARS-CoV-2 infection due to the Omicron variant. Considering the enormous amounts of COVID-19 patients, in particular after the emergence of the Omicron variant, more studies might be necessary to elucidate virulence and pathophysiology regarding different SARS-CoV-2 variants in order to establish optimal diagnostic and therapeutic strategies in each pandemic/endemic period of COVID-19.

## Figures and Tables

**Figure 1 viruses-15-01304-f001:**
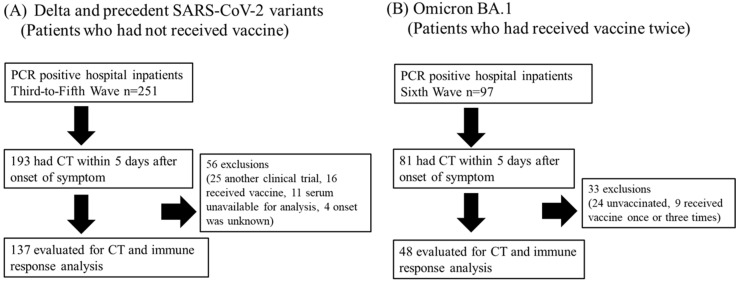
Flow chart illustrating the distribution of participants in this study.

**Figure 2 viruses-15-01304-f002:**
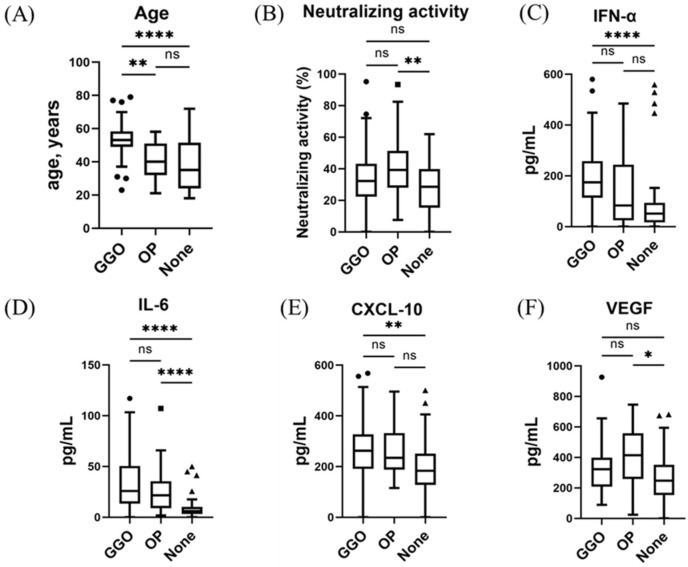
Serum immune indicator levels and associations with dominant CT patterns in the early phase of infection with SARS-CoV-2 Delta and precedent variants: (**A**) age, (**B**) neutralizing activity (NT; % inhibition), (**C**) IFN-α, (**D**) IL-6, (**E**) CXCL10, and (**F**) VEGF. No subjects had received a vaccine against SARS-CoV-2 before admission. Each biomarker level was evaluated at the time of hospital admission (within five days after symptom onset). Data are presented using Tukey boxplots and individual values. ns, not significant; * *p* < 0.05; ** *p* < 0.005; **** *p* < 0.0001.

**Figure 3 viruses-15-01304-f003:**
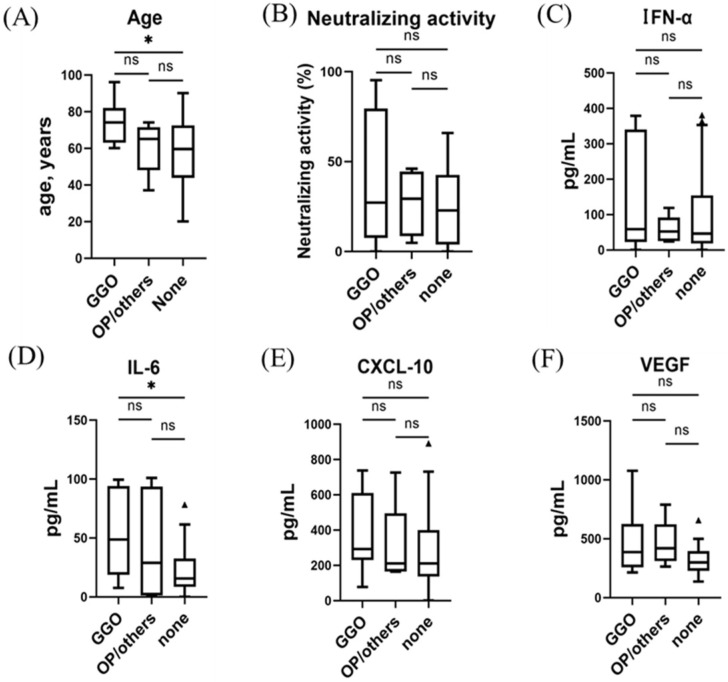
Serum immune indicator levels and associations with dominant CT patterns in the early phase of infection with the SARS-CoV-2 Omicron variant: (**A**) age, (**B**) neutralizing activity (NT; % inhibition), (**C**) IFN-α, (**D**) IL-6, (**E**) CXCL10, and (**F**) VEGF. All participants had received an mRNA vaccine against SARS-CoV-2 (wild-type) more than two weeks before infection. Each biomarker level was evaluated at the time of hospital admission (within five days after symptom onset). Data are presented using Tukey boxplots and individual values. ns, not significant; * *p* < 0.05.

**Figure 4 viruses-15-01304-f004:**
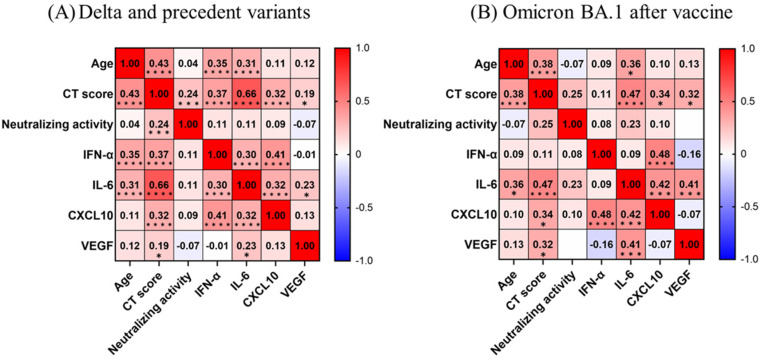
Correlation matrix of biomarkers in participants with early-phase SARS-CoV-2 infection (within five days after symptom onset). (**A**) Unvaccinated participants with COVID-19 from precedent variants to Omicron (during the third to fifth waves). (**B**) Vaccinated participants with COVID-19 from the Omicron variant (during the sixth wave). Spearman correlation coefficients are plotted. Cells are colored according to the strength and trend of correlations (shades of red = positive correlations, shades of blue = negative correlations). * *p* < 0.05; *** *p* < 0.001; **** *p* < 0.0001.

**Figure 5 viruses-15-01304-f005:**
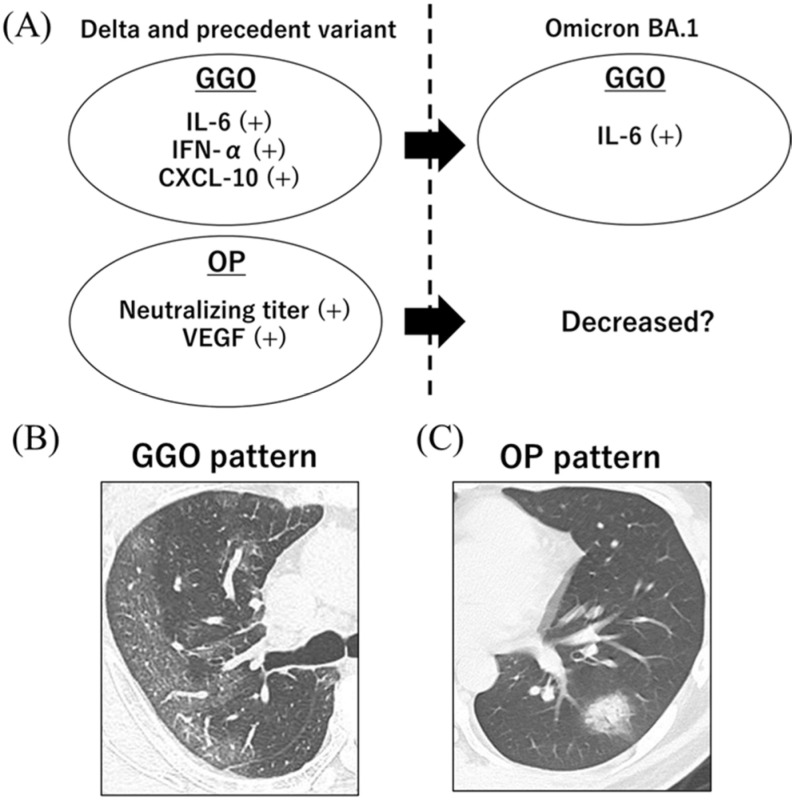
Landscape of immune response and dominant CT patterns during the early phase of SARS-CoV-2 infection (**A**). During the third to fifth waves, where Delta and precedent variants were the dominant variants, serum IL-6, IFN-α, and CXCL10 levels were significantly higher in unvaccinated participants with GGO patterns, and serum IL-6, IFN-α, and CXCL10 levels were significantly higher in those with OP patterns when compared to those without pulmonary lesions. During the sixth wave, where the dominant variant was Omicron BA.1, only serum IL-6 levels were significantly higher in vaccinated participants with GGO patterns when compared to those without pulmonary lesions. Notably, the presence of OP patterns was significantly decreased in participants with the Omicron variant. Chest CT images showing the dominant patterns in patients with early COVID-19; (**B**) GGO pattern, (**C**) OP pattern.

**Table 1 viruses-15-01304-t001:** Clinical features of patients with COVID-19 in this study.

	Delta and Precedent Variants (n = 137)	Omicron BA.1 after Vaccine (n = 48)	*p*-Value
Age, years	49 (32–54)	63 (51–74)	<0.001
Sex; number of male/female	79/58	31/17	0.401
Underlying disease			
None	74 (54)	14 (29)	0.003
Hypertension	25 (18)	20 (42)	0.001
Diabetes mellitus	8 (6)	8 (17)	0.046
Body mass index (kg/m^2^)	22.5 (21–25]	25.4 (23–28]	<0.001
Initial nasopharyngeal viral load (log copies/μL)	4.8 (3.8–5.6]	4.4 (4.0–5.0]	0.112
RNAemia	31 (23)	5 (10)	0.104
Laboratory data			
Neutrophil-to-lymphocyte ratio	2.6 (1.6–4.0)	2.9 (1.9–5.8]	0.055
LDH (IU/L)	195 (173–217]	180 (166–209]	0.338
CRP (mg/dL)	0.6 (0.2–1.7]	1.3 (0.8–3.3]	0.046
D-dimer (ng/mL)	0.7 (0.6–0.9]	0.8 (0.6–1.5]	0.260
Respiratory failure	33 (24)	8 (17)	0.389
Duration of oxygen therapy (days)	7 (4–11]	9.5 (3–30]	0.153
IPPV	2	2	0.594
Nasal high flow	4	0	0.535
Death within 30 days after onset	0 (0%)	0 (0%)	—

Continuous variables are reported as medians (interquartile range (IQR) 25–75]. Categorical variables are reported as numbers (percentages). IPPV: intermittent positive pressure ventilation; ‘—’ indicates that the data were not applicable for comparison.

**Table 2 viruses-15-01304-t002:** Radiological features of patients with COVID-19 in this study.

	Delta and Precedent Variants (n = 137)	Omicron BA.1 after Vaccine (n = 48)	*p*-Value
CT-evaluated day from clinical onset	3 (2–4]	2 (1–3]	0.003
Absence of abnormal pulmonary lesions	61 (45)	32 (67)	0.008
Semi-quantitative CT score	1.5 (0–6]	0 [0,1]	0.041
Dominant CT pattern			
GGOs	46 (34)	11 (23)	0.169
OP	27 (20)	1 (2)	0.007
Others	3 (2)	4 (8)	0.139
Accompanied CT manifestation			
Reversed halo shadow	7 (5)	2 (4)	1.000
Curvilinear shadow	15 (11)	5 (10)	1.000
Bronchovascular bundle thickening	17 (12)	6 (13)	1.000
Traction bronchial dilation	3 (2)	2 (4)	0.834

Continuous variables are reported as medians (interquartile range (IQR) 25–75]. Categorical variables are reported as numbers (percentages).

## Data Availability

All data generated or analyzed during this study are included in this published article and its Appendix A files.

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
