# Peer review of "Dominant CT Patterns and Immune Responses during the Early Infection Phases of Different SARS-CoV-2 Variants"

_viruses, 2023, doi:10.3390/v15061304_

Round 1

Reviewer 1 Report 

General feed-back

This  study, part of the Toyama University COVID-19 Cohort Study (Japan), examined  the clinical, epidemiological, radiological, and microbiological features of patients diagnosed with COVID-19 between December 2020 and April 2022 in Japan, covering four major COVID-19 waves.  

Study participants included those not immunized during the third through fifth waves (Delta and precedent) and  those immunized twice at least 2 weeks before admission-to hospital during the sixth wave (Omicron BA.1 after vaccine) 

The final number of study subjects was patients hospitalized was 137 patients infected by pre-omicron variants versus 48 vaccinated patients infected by Omicron. 

Main findings

Early COVID-19 immune response differed between patients with ground glass opacities (GGO) or organized pneumonia (OP) for infection with Delta and precedent variants. Serum levels of IFN-α and CXCL10 strongly correlated with GGO, whereas neutralizing activity and VEGF were associated with OP. Patients with GGO as dominant CT pattern were significantly older than those with OP as dominant feature

Whilst OP prevalence significantly decreased with Omicron, IL-6 strongly correlated with CT score. Regardless of the variant. 

Specific comments

Lines 42-44: when mentioning the high infectiousness of Omicron with respect to previous variants citations are needed . For instance these two studies (PMID: 36016284  PMID: 35215930) provide changing epidemiological figures and vaccine effectiveness in the same population of routinely/mandatorily screened health care workers (HCWs), before and after Omicron, with non-occupational (asymptomatic or mild-moderate) SARS-Cov-2 infection increasing dramatically during Omicron, despite high vaccination coverage. 

Methods

Did the authors excluded/considered also incidental COVID-19 infections? 

Why did the authors pick up only unvaccinated individuals for pre-omicron variants and only vaccinees during Omicron time? It would be interested to seethe impact of vaccination in pre-omicron strains and the impact of omicron on unvaccinated (possibly also never infected before) individuals. 

The authors could also perform some regression (not just correlation) analysis to assess risk factors and relative risk for CT-score, OP or GGO or increased serum levels of main markers (e.g. IL-6 or CXCL10) 

Lines 321-22:This is because the history of vaccination was less associated with the early immune response and CT patterns in participants from the sixth wave”. Here the authors should justify this statement. Is this a supposition or is it sustained by evidence? In which case a citation is needed

Lines 328-39: again, this statement needs backing citation. 

Line 346-47: Whilst it is true that Tocilizumab targets IL-6, antivirals as Molnupiravir or Paxlovid proved equally effective against severe COVID-19 as monoclonal antibodies. Moreover, oral antivirals are also more practical di administered than monoclonal antibodies i.v. Antivirals also have the secondary benefit of reducing the viral shedding time, according to recent real time research (https://doi.org/10.3390/ph16050721).

Reviewer 2 Report

I read with great interest the paper. I find it well wrote and with good idea research. Below my suggestions:

1. Introduction: please add cumulative case of SARS CoV2 at the day of resubmission and the impact of COVID-19 pandemic also on other diseases increasing diagnostic delay and worste outcome 

2.Methods and results: in my opinion, true well presented. Clear and efficacy. also the table and figure are well done.

3. Discussion: when we discuss on SARS CoV2 we cannot avoid talking about long covid. Please add some considerations on Long Covid and also on therapies and strategies to improve long covid (see Incidence of long COVID-19 in people with previous SARS-Cov2 infection: a systematic review and meta-analysis of 120,970 patients. Intern Emerg Med. 2022 Nov 30:1–9. doi: 10.1007/s11739-022-03164-w.  and  Interventions for Improving Long COVID-19 Symptomatology: A Systematic Review. Viruses. 2022 Aug 24;14(9):1863. doi: 10.3390/v14091863. )

Conclusion: give some public health consideration that came from your great paper. 

Round 2

Reviewer 1 Report

The authors addressed most comments.

However, there is an outstanding critical point in discussion which should be explained/addressed, as I believe it is around the main message of this research study (Lines 336-347):

Line 341: “increased resistance to the innate immune defense”… “immune defences” is an inappropriate term as it refers to IFN.  Please refer to “aspecific physiological defences of upper airways” (Ph, mucus, enzimes, et.).  PMID: 36839483, PMID: 36432693

Lines 342-44: “Thereafter, owing to inefficient replication in human lungs, [10, 11], proliferation or invasion of the Omicron variant may be highly suppressed when the immune response (including the early humoral immune response) is effectively induced”… this sentence is in contradiction with the above statement “ intrinsic viral factors rather than the impact of the vaccine predominantly affected the CT patterns of participants with the Omicron variant”. Vaccine in fact confer humoral response. 

Minor point

Lines 195-196: Amongst, one patient developed COVID-19 due to Delta variant in hospital (incidental COVID-19) due to Delta variant…. “Delta variant” repeated twice
